# STXBP1 Syndrome: Biotechnological Advances, Challenges, and Perspectives in Gene Therapy, Experimental Models, and Translational Research

**DOI:** 10.3390/biotech14010011

**Published:** 2025-02-20

**Authors:** Silvestre Ruano-Rodríguez, Mar Navarro-Alonso, Benito Domínguez-Velasco, Manuel Álvarez-Dolado, Francisco J. Esteban

**Affiliations:** 1Andalusian Center for Molecular Biology and Regenerative Medicine (CABIMER), CSIC-US-JA-UPO, Américo Vespuccio Avenue 24, Cartuja Scientific and Technological Park, 41092 Seville, Spain; silvestre.ruano@cabimer.es (S.R.-R.); mar.navarro@cabimer.es (M.N.-A.); benito.dominguez@cabimer.es (B.D.-V.); 2Systems Biology Unit, Department of Experimental Biology, University of Jaén, Campus Las Lagunillas s/n, 23071 Jaén, Spain

**Keywords:** antisense oligonucleotides, bioinformatics, epilepsy, gene and cell therapy, genomics, haploinsufficiency, induced pluripotent stem cells, infantile epileptic encephalopathy, murine models, personalized medicine, STXBP1

## Abstract

STXBP1 syndrome is a severe early-onset epileptic encephalopathy characterized by developmental delay and intellectual disability. This review addresses key challenges in STXBP1 syndrome research, focusing on advanced therapeutic approaches and experimental models. We explore gene therapy strategies, including CRISPR-Cas9, adeno-associated viral (AAV) vectors, and RNA therapies such as antisense oligonucleotides (ASOs), aimed at correcting STXBP1 genetic dysfunctions. This review presents in vivo and in vitro models, highlighting their contributions to understanding disease mechanisms. Additionally, we provide a proposal for a detailed bioinformatic analysis of a Spanish cohort of 41 individuals with STXBP1-related disorders, offering insights into specific mutations and their biological implications. Clinical and translational perspectives are discussed, emphasizing the potential of personalized medicine approaches. Future research directions and key challenges are outlined, including the identification of STXBP1 interactors, unexplored molecular pathways, and the need for clinically useful biomarkers. This comprehensive review underscores the complexity of STXBP1-related infantile epileptic encephalopathy and opens new avenues for advancing the understanding and treatment of this heterogeneous disease.

## 1. Introduction

STXBP1 syndrome (syntaxin-binding protein 1 syndrome) is a severe infantile epileptic encephalopathy (EIEE4, OMIM: #612164) [1] caused by mutations in the STXBP1 gene, which encodes the syntaxin-binding protein 1, also known as Munc18-1 (UniProt: P61764) [2,3,4,5]. This presynaptic protein plays a crucial role in intracellular membrane trafficking, vesicle secretion mechanisms, and particularly in neurotransmitter release [3,6,7,8]. Located in the chromosomal region 9q34.1, STXBP1 mutations result in a wide range of mostly de novo variants associated with diverse phenotypic comorbidities resembling Ohtahara (OMIM:#308350) [1], Dravet (OMIM:#607208) [1], West (EIEE5, OMIM:#613477) [1], or Lennox–Gastaut (OMIM:#301058) syndromes, among others [4]. These syndromes are characterized by their onset in infancy, with a highly variable etiology and natural history. While seizures are the hallmark symptom of all EIEE syndromes, often accompanied by progressive cognitive impairment, these disorders exhibit significant diversity in age of onset, severity, seizure types, EEG patterns, associated symptoms, and overall prognosis [9].

The clinical manifestations of STXBP1 syndrome encompass a broad spectrum of neurological and developmental challenges. These include severe intellectual disability in most individuals, multiple seizure types (e.g., tonic–clonic, myoclonic, infantile spasms), autism spectrum characteristics, tremors, ataxia, aggressive behaviors in some cases, and communication issues [4,10,11]. The onset of symptoms typically occurs within the first few days of life or by one year of age, with the timing and severity of different clinical manifestations varying based on factors such as mutation type, seizure onset age, and seizure type [3,12,13].

The heightened susceptibility to seizures in infants with STXBP1 syndrome may be attributed to the immature state of their brains, characterized by unique neural network properties and continuous modifications during development [14]. Electroencephalogram (EEG) analysis provides crucial information about the disease, with studies indicating that patients with pathogenic STXBP1 mutations often exhibit a predominantly inhibitory neural state, mainly associated with GABAergic signaling. This imbalance between excitation and inhibition is thought to be a primary cause of seizures in affected infants [15,16].

Animal models have provided valuable insights into the role of STXBP1 in neuronal function. Homozygous null murine models demonstrate that the complete absence of the Munc18-1 protein results in a lack of synaptic activity and transmission [4,5,17,18,19]. These findings underscore the critical importance of STXBP1 in maintaining normal neuronal function and synaptic communication.

Despite the growing understanding of STXBP1 syndrome, its prevalence may be underestimated due to various factors, including symptom overlap with other neurological conditions, the complex and costly diagnostic process, limited knowledge of disease progression, and lack of widespread access to genetic testing [20]. The main pathogenic mechanism of the syndrome is STXBP1 haploinsufficiency, where many mutations lead to protein aggregation, degradation, or mRNA loss, likely due to protein instability.

Currently, there is no effective treatment without severe side effects for STXBP1 syndrome. Anticonvulsant drugs are used to control epileptic seizures, but a significant percentage of patients show resistance to these or require a combination of high-dose drugs, sometimes leading to devastating side effects [21]. The limitations in current treatment options highlight the urgent need for new therapeutic approaches, which require a deeper understanding of encephalopathy and its progression, identification of regulatory genes and biomarkers, and development of accurate disease models.

This review aims to address the key challenges in STXBP1 syndrome research, exploring advanced therapeutic approaches, experimental models, and future clinical perspectives. By synthesizing current knowledge and highlighting recent advancements, we hope to contribute to the ongoing efforts to improve diagnosis, treatment, and quality of life for individuals affected by this complex neurological disorder.

## 2. Pathogenic Mechanism and Types of Existing Mutations in STXBP1 Syndrome

The STXBP1 syndrome is primarily caused by haploinsufficiency, where mutations lead to reduced levels of functional Munc18-1 protein, impairing synaptic vesicle exocytosis and neurotransmitter release [3,6,7,8]. Several types of mutations have been identified in the STXBP1 gene, including nonsense mutations (~50% of cases), frameshift mutations, splice site mutations, missense mutations, and deletions (intragenic, whole gene, and multi-gene) [5,20].

Nonsense and frameshift mutations often result in truncated proteins that are unstable and rapidly degraded. Splice site mutations can lead to aberrant mRNA processing, while missense mutations may affect protein folding, stability, or interactions with binding partners [21,22,23]. The Munc18-1 protein consists of three domains, with mutations distributed across all regions. No specific domain appears to be particularly susceptible or spared from mutations. Some recurrent variants, such as p.Arg406His, p.Arg406Cys, and p.Arg292His, are considered mutational hotspots [2,24,25] (Figure 1).

Despite the identification of over 300 pathogenic variants, establishing clear genotype–phenotype correlations remains challenging due to the high phenotypic variability among patients, even those sharing the same mutation. However, some recurrent variants have been associated with specific clinical features. For instance, the p.Arg406Cys/His variant is linked to burst suppression EEG patterns, spastic tetraplegia, and inability to walk [2,26].

While haploinsufficiency is the predominant pathogenic mechanism, some variants exhibit different effects. A small percentage of missense variants may lead to protein aggregation, potentially trapping functional copies and exacerbating STXBP1 loss of function. Rare variants, such as L446F, have been reported to cause a gain-of-function phenotype in synaptic transmission, suggesting a possible dominant-negative mechanism [27,28].

Understanding these diverse mutational mechanisms is crucial for developing targeted therapies and improving patient outcomes in STXBP1 syndrome. The complexity of the genetic landscape underscores the need for comprehensive genetic testing and individualized approaches to treatment and management.

The STXBP1 gene encodes the Munc-18 protein, a Sec1-like protein with a single structural domain but composed of three functional subdomains: Domain 1, Domain 2, and Domain 3 (further subdivided into 3A and 3B). The model used to extract the structural information of the protein is PDB ID: 3PUJ from Protein Data Bank (https://www.rcsb.org/structure/3PUJ) accessed on 10 February 2025 [29,30]. This distribution suggests the presence of mutational “hotspots” in the STXBP1 gene structure. All depicted variants are pathogenic but vary in their degree of pathogenicity. The R292H (orange), R406C (red), and R406H (red) variants are loss-of-function mutations causing haploinsufficiency, while the L446F (pink) variant is atypical, representing a gain-of-function mutation with a dominant-negative pathogenic mechanism. An image was created using BioRender January 2025 version, based on the article’s image by Uddin et al. [24,31] and rendered with Chimera-X [32].

## 3. Gene and Cell Therapy: Approaches and Challenges

Gene and cell therapy represents the most promising strategies for advancing the treatment of STXBP1 syndrome. Mutations in STXBP1 lead to a loss of function in the Munc18-1 protein. The initial goal of gene therapy is to restore normal protein levels in neurons, thus compensating for the effect of haploinsufficiency [5,19,23]. Current approaches in gene therapy include the administration of functional copies of the STXBP1 gene using viral vectors such as adeno-associated viruses (AAV), gene-editing tools like CRISPR-Cas9, and antisense oligonucleotides (ASO). Each therapeutic strategy presents technical and safety challenges that still limit their clinical application [21].

Studies using the CRISPR/Cas9 mechanism have been developed in order to understand the pathophysiology of the syndrome and to develop a targeted therapy [21]. Ichise and colleagues [33] generated an isogenic control line by repairing a pathogenic heterozygous variant in iPSCs derived from patients with STXBP1 syndrome. They also differentiated iPSCs into GABAergic neurons through the transient expression of the transcription factors ASCL1 and DLX2 [34]. In this way, a GABAergic neuronal model derived from STXBP1 patients was generated and compared with isogenic control neurons. This model reinforced the idea that the GABAergic system is a crucial factor in STXBP1 pathogenesis, along with aberrant expression in genes associated with epilepsy, neurological development, and neurodegeneration in iPSC-derived neurons from STXBP1 patients [34,35].

A novel gene therapy yielding positive results in various research groups studying other epileptic encephalopathies and channelopathies, such as Dravet syndrome, is the use of AAV vectors for gene transfer (adeno-associated viruses). This approach has also been started with STXBP1, aiming to counter haploinsufficiency by replacing or regulating the gene through the addition of a completely functional new copy. One example is the construct created by Chen and colleagues [36], which reversed the abnormal hindlimb grip, reduced the frequency of myoclonic seizures, and addressed cognitive comorbidities such as anxiety-like behaviors and nest-building restoration [4]. Moreover, the use of AAV capsids containing ASOs inside could improve brain biodistribution [21].

RNA therapies via antisense oligonucleotides (ASOs) have emerged in recent years as a promising strategy for diseases caused by mutations, such as STXBP1 [37,38]. These synthetic RNA are designed to selectively bind to specific mRNA regions and modulate their expression or processing, even correcting genetic alterations at the post-transcriptional level. Thus, targeting the STXBP1 gene offers an innovative approach that could restore expression balance and STXBP1 protein function. When designing ASOs to enhance protein expression or modulate splicing to prevent pathogenic mutations, the goal is to partially reverse the functional deficit in affected neurons. This approach is promising because ASOs can be administered directly into the central nervous system, and being chemically modifiable, they can be stabilized for prolonged and specific action. However, implementing this approach in the clinic is complex and requires a deep understanding of STXBP1 mutational variability and preclinical studies in models that allow efficacy and safety evaluation in the STXBP1 syndrome context [39,40]. This approach has already been implemented in Dravet Syndrome [41]

In addition to ASOs, long non-coding antisense RNAs can be used to increase mRNA translation and, in our case, STXBP1 protein levels [42]. Additionally, considering that most STXBP1 disorders are associated with haploinsufficiency, physiological microRNAs can target STXBP1 transcription to significantly reduce transcription, or increase it if the therapy targets the microRNA [4].

Thus, one of the approaches using ASOs as gene therapy for STXBP1 syndrome aims to elevate cellular levels of functional protein by ASOs competing with repressor miRNAs to bind mRNA sites and avoid its degradation, leading to higher levels of mature and functional mRNAs that could produce more protein. Despite similar therapeutic approaches showing promising results in other encephalopathies such as Dravet syndrome [41] (ClinicalTrials ID: NCT04740476), only patients with loss of function mutations could benefit from it [4]. Even though some ongoing studies about STXBP1 haploinsufficiency treatment with transcription enhancing ASOs have yet to be published. A small percentage of STXBP1 missense variants apparently have the propensity to aggregate, trapping correctly formed copies and impairing STXBP1 loss of function [43]. In this situation, a mixed strategy using specific ASOs that bind to mutant mRNA leading to its degradation and transcription enhancing ASOs could be the most suitable approach.

Although AAVs have been widely used for gene therapy delivery, exosomal vesicles (EVs) are a promising vehicle for ASOs administration that has been proved effective in different diseases such as Parkinson’s disease [44]. These particles can be customized to target specific regions or cell types, leading to highly specific treatments, avoiding virus-based delivery systems and minimizing off-target effects.

Cell therapy is another approach that has been investigated as a potential treatment. Previous works conducted on temporal lobe epilepsy (ClinicalTrials ID: NCT06422923) have shown the potential role of interneuron grafts in many diseases, as autism, schizophrenia, neurodegenerative diseases, and epilepsy [45]. GABAergic interneurons are essential in keeping the correct excitatory–inhibitory balance and, given the reported alteration of these neurons in STXBP1 mouse models, they are a main therapeutic target to be restored in this syndrome and other epileptic conditions. In this way, using a pilocarpine mouse model of pharmacoresistant epilepsy, Hunt et al. [46] found a therapeutic effect in seizures and behavioral impairment after the transplantation of inhibitory precursor cells from the Medial Ganglionic Eminence (MGE). This region is the embryonic brain area where cortical and hippocampal interneurons are born during normal development [47]. Our group has also extensive experience in the use of MGE-derived interneuron precursors for cell therapy. We have shown that MGE-derived progenitors differentiate into GABAergic interneurons when they are transplanted into the normal neonatal and adult mouse brain [48,49]. Unlike most other neuronal precursors, MGE cells migrate long distances (3–4 mm.), integrating widely in the cerebral cortex, hippocampus, and striatum. They differentiate, acquiring characteristic morphologies and molecular markers of mature GABAergic interneurons. We showed their integrations into the host circuitry and electrophysiological functionality, presenting typical action potentials and membrane properties of mature GABAergic interneurons, and modifying the general inhibitory tone in the grafted zones of the host [48,49]. These characteristics strongly suggest their suitability for a therapeutic application in Dravet syndrome (DS). Their naturally occurring differentiation to mature GABAergic neurons, and their wide migration through the affected areas, would make them ideal to replace the altered GABAergic function observed in STXBP1 mouse models, or, at least, as a direct source of GABA that may prevent seizures. Supporting this hypothesis, we demonstrated that grafts of these precursors replace physically and functionally a specific deficit in hippocampal interneurons that lead to higher susceptibility to seizures in adult mice [49]. In addition, grafts of these precursors present a robust anticonvulsant activity tested in the maximal electroshock seizure assay, avoiding seizure generalization and reducing significantly the mortality rate as a consequence of induced seizures [50]. Finally, these cells are extremely stable and safe, since survival of the cells and no tumor formation has been reported more than one year after their transplant in a mouse brain. Moreover, we transplanted these cells into a mouse model of Alzheimer’s disease (AD) that presents deficits in Nav1.1 expression, as occurs in DS. The results [51] show the rescue of cognitive alterations and brain rhythms in this model after the transplants. The analysis of the EEG wave spectrum indicates a normalization of cerebral rhythmogenesis in these animals, which leads to a reduction in their hyperactive-anxiety behaviors, and the recovery of cognitive levels.

To implement the therapeutic application of MGE-derived interneurons in the clinic, a permanent source of these cells is necessary. This could be achieved by using iPSC and controlling their differentiation with protocols already established. It has been recently published that MGE-like interneurons obtained from embryonic stem cells mediate positive effects in the reduction in epileptic seizures in a model of temporal lobe epilepsy [52]. This approach has the great advantage of having an unlimited source of cells for transplantation and they can be generated from the patient’s own fibroblast, preventing immunological rejection.

As a whole, these results strongly suggest the suitability of MGE-derived neuronal precursors for cell-based therapies, and open a new avenue for the treatment of disorders related to deficits in interneurons, as is the case of STXBP1 syndrome. We are currently working to confirm if this cell-based therapy is preclinically feasible in mouse models of STXBP1.

Some of these therapies, in different development stages, will hopefully reach the clinical trial stage, while others will be discarded. It is also expected that new therapeutic approaches will emerge in the coming years thanks to the high visibility that pediatric epileptic encephalopathies are currently receiving, especially due to the solid work by foundations. However, any personalized therapy approach that proves effective could address a significant part, or possibly all in the best cases, of the clinical manifestations linked to STXBP1 deficiency encephalopathy.

## 4. Experimental Models Used in STXBP1 Syndrome Research

Experimental models are key to understanding pathological mechanisms and exploring potential therapeutic interventions for STXBP1 syndrome. Particularly, mouse models are the most widely used and have enabled the analysis of numerous mutations’ effects on neuronal development and function, providing various insights into synaptic and neuronal excitability alterations. These models include knockout (KO), knock-in (KI), and humanized mice, accurately replicating disease mutations. This allows us to observe phenotypic manifestations similar to those observed in patients. Other models involve the use of KO zebrafish for STXBP1 homologs, STXBP1A and STXBP1B [5,53]; *Drosophila melanogaster* (KO of Rop, the orthologous gene in this species) [5,54,55]; and *C. elegans* (KO of Unc-18) [5,56,57].

In Droshophila, inactivating the STXBP1 homolog (Rop) causes embryonic lethality. In the case of heterozygous null mutations, organisms remain viable with normal survival, but they exhibit reduced synaptic release [5,54,55]. In zebrafish, removing the *stxbp1a* and *stxbp1b* homologs results in immobility and death of individuals, and a seizure-like phenotype, respectively, accurately reflecting STXBP1 disorder [53]. In *C. elegans*, the Unc-18 homolog’s elimination results in severe paralysis and highly depressed neurotransmitter release [56]. A distinctive feature of *C. elegans* compared to zebrafish and fruit flies is that KO organisms remain viable [5].

Moreover, using non-animal models such as patient-derived induced pluripotent stem cells (iPSCs) [58] or human brain organoids [59,60] facilitates the study of variants associated with various neurodevelopmental disorders in a human model, promoting the development of new targeted therapies and more accurate preclinical evaluations.

As for mouse models, humanized mice are particularly novel. In the case of STXBP1, these models have been less successful due to reduced viability compared to other humanized mice in genes such as SYNGAP1 (Syngap1 Hu/Hu) when using biallelic humanization. These KI models are non-conditional and include the entire gene locus, covering the promoter, UTRs, proximal enhancers, coding sequence, and intronic regions. This approach enables the inclusion of the gene of interest’s antisense transcription in the humanized region, enabling multiple therapeutic approaches targeting this element. Hybrid animals (Hu/+) and littermate WT (+/+) are also generated, allowing for routine genotyping. Furthermore, humanized mice could be crossed with haploinsufficient (+/−) mice to evaluate phenotypic rescue after a therapeutic intervention using compounds targeting human genes. In both cases, the resulting mice are viable and can be particularly useful for the scientific community [61]. However, a limitation common in these models is that humanizing a gene may affect the expression of other genes, as they share regions like the promoter [59] or enhancers, which can act even over kilobase distances [62]. This issue is observed in the SYNGAP1 example [61].

KO models are widely used and have been crucial for studying the impact of Munc18-1 protein loss of function in the nervous system [63]. These KO mice can be global KOs or conditional KOs. Global KOs have been instrumental in demonstrating that mice with complete STXBP1 deletions do not survive, underscoring Munc18-1 protein’s importance in synaptic transmission in both excitatory and inhibitory neurons and highlighting its necessity in the central nervous system. Meanwhile, conditional KO mice [64] allow the study of specific effects while avoiding early lethality by restricting deletion to particular brain regions or cell types. One significant example is GABAergic neuron-specific KOs, revealing that their dysfunction leads to synaptic dysfunction and subsequent neuronal hyperexcitability, exacerbating seizure phenotypes, and behavioral and cognitive alterations [65]. Other conditional KO studies focus on glutamatergic neurons [66], with notable effects on synaptic plasticity, learning difficulties, memory, autistic traits, and other functional and cognitive development manifestations.

## 5. Potential Strategy for the Bioinformatic Analysis of Genomic Data from Our Cohort of Patients

In this section, we outline a comprehensive plan to be implemented in the near future for the bioinformatic analysis of genomic data from a cohort of Spanish patients with STXBP1 syndrome. This cohort represents a significant portion of the identified cases in Spain, making it a valuable resource for understanding the genetic landscape of the disorder in this population. By analyzing this cohort, we will aim to uncover genotype–phenotype correlations, identify potential molecular comorbidities, and contribute to the development of personalized medicine approaches for STXBP1 syndrome. The following paragraphs will detail our methodology, including sample collection, sequencing techniques, and bioinformatic analysis procedures, which have been carefully designed to extract meaningful insights from this unique dataset.

Our patient cohort consists of 41 individuals diagnosed with STXBP1 syndrome and 24 siblings who serve as controls, with a broad age range from 3 to 27 years and a balanced gender distribution. Thus, this cohort represent over 60% of the identified cases of the disease in Spain. Clinical data were provided by participating families and collected by pediatric neurologists specializing in the management of epileptic encephalopathies (EIEEs). Besides the medical history, a standardized questionnaire was created to collect the most relevant information, which will later be integrated into the computational analysis of the samples. This questionnaire includes factors such as demographic data, comorbidities, response to anticonvulsant medications, cognitive effects, etc.

Sample collection was carried out through peripheral blood extractions from patients, thanks to collaboration with the National STXBP1 Syndrome Association. Consequently, in addition to DNA information, RNA and serum samples are available from patients with STXBP1 syndrome. The samples were sequenced in parallel using short-read WGS. Data libraries were prepared with Illumina preparation kits as recommended by the manufacturer, specifically using the Nextera DNA Prep Kit for WGS.

Bioinformatic processing of short-read WGS data will be conducted through the Picard-GATK pipeline on the Hercules Supercomputing Cluster at CICA (Centro Informático Científico de Andalucía) (Figure 2). Once the raw sample reads are obtained, they will undergo quality control using FastQC v0.11.5 to identify any sequencing errors or lower-quality data and enhance the efficiency of subsequent analyses. To eliminate most noise, they will be processed with Trimmomatic v.0.39, a Java-based tool (Java v17.0.6), performing quality-based trimming and removing Nextera adapters. Cleaned reads will be subjected to a second quality control check before alignment to the curated reference genome GRCh38/hg38, downloaded from Ensembl, using BWA-MEM v0.7.17. It is crucial to index the genome beforehand to improve efficiency and reduce computational cost. The reads will be then sorted by coordinates to streamline further analysis, and alignment success will be verified. These steps will be conducted using SAMtools v1.17, a highly versatile bioinformatics tool for sequence manipulation and analysis. In nearly all patient samples, alignment exceeds 99%, confirming the reliability of the analysis. Sample duplicates will be marked using Picard v2.25.0, also Java-dependent, for later removal to further refine the reads for Variant Calling analysis. This same software will be used to add and replace read groups, facilitating the subsequent analysis and ensuring that samples are organized and labeled correctly.

Next, all steps will be carried out with GATK v4.5.0.0. The first task will be to recalibrate the bases in the reads to improve the accuracy of variant calling and adjust the quality scores, correcting for any systematic errors that occur during sequencing. This process will use the BaseRecalibrator and ApplyBQSR functions. With this recalibration complete, variant calling will be conducted with GATK HaplotypeCaller, identifying indels and SNPs based on the 1000-GENOMES database. This computationally intensive process will be followed by variant filtering through several parameters to yield two files with indels and SNPs separated. The SelectVariants function in GATK will be used for this step. Once filtered, we will annotate the variants with GATK’s Funcotator function to obtain our final results. However, it will first be necessary to download the variant package and organize it as specified on the GATK website (https://gatk.broadinstitute.org/), accessed on 10 February 2025. This downloaded data package will include variant information from numerous databases such as ClinVar, gnomAD, and Gencode, among others.

## 6. Clinical and Translational Perspectives of STXBP1 Syndrome

Currently, no standardized treatment strategy exists for children with STXBP1 disorder. Treatment approaches primarily focus on reducing seizure frequency, though results are generally limited. In many cases, patients either show resistance to anticonvulsant medications or revert to frequent seizures after a period of reduced epileptic activity. Side effects can also have devastating impacts. Common anticonvulsant medications (ASMs) used include levetiracetam, clobazam, phenobarbital, topiramate, sodium valproate, vigabatrin, and, in some cases, hormonal treatments such as adrenocorticotropic hormone (ACTH) [69]. Another non-pharmacological option is the ketogenic diet [70].

Medication regimens change significantly with age, with some patients trying up to eight different treatments over their lifetime. Typically, phenobarbital and levetiracetam are used early in life, alongside ACTH, while clobazam and ketogenic diets are often introduced in adolescence and adulthood. Although each medication reduces seizures in some patients, efficacy varies based on patient age. Developing a quantifiable ASM response assessment could create a foundation for evaluating new treatments [25].

In the near future, drug repurposing will be crucial to finding alternatives to existing clinical treatments. Exploring personalized medicine strategies for STXBP1 epileptic encephalopathy will be essential to adapt therapies to the specific genotypic and phenotypic characteristics of each patient, despite the challenges. Emerging therapies are showing promise, and the bio-pharmaceutical industry’s growing interest in targeted intervention research is a significant step forward [21].

Identifying new biomarkers to enable early diagnosis and monitor disease progression through prospective studies, which remain underexplored, will also be critical [71]. Addressing the impact of the condition on patients’ and caregivers’ quality of life is also paramount, considering the emotional and psychological burden it places on families. Early diagnosis is crucial to avoid unnecessary testing, guide families in handling challenges, and provide resources [17].

The current distribution of affected patients suggests that the prevalence of STXBP1 syndrome may be underestimated due to limited access to diagnostic testing, along with delays in obtaining a definitive diagnosis. Optimal management of these cases requires multidisciplinary teams, including neurology, psychology, and occupational therapy specialists, as well as caregiver support programs and networks that offer resources and guidance [17]. Quality of life will be a key measure of success for therapeutic interventions, aiming to address both the physical and psychosocial aspects of the disorder. All the above-mentioned key factors have been compiled in Figure 3.

## 7. Challenges and Future Directions in Research

The pace of advances in personalized treatments for STXBP1 suggests a hopeful future; however, significant questions remain. To continue this progress, it will be essential to better understand the goals of personalized therapy for STXBP1 syndrome [22]. Although reducing seizure frequency is an indicator of success, multiple comorbidities still affect quality of life, especially motor and cognitive symptoms. For this reason, many studies aim not only to reduce seizures but also to fully elucidate some unknown molecular mechanisms of STXBP1 pathology, such as its role in other neuronal functions like synaptic plasticity, the Munc18-1 interactome [72], or its possible role in other nervous system cells such as glial cells (in processes of communication and neuronal support). Identifying genes involved in STXBP1 regulation and metabolic alterations in childhood epileptic encephalopathies will be fundamental to finding relevant molecular pathways [73].

In recent years, animal models and in vitro assays have been on the rise. Mouse models will be critical for analyzing disease progression over time. These models are excellent for understanding disease mechanisms and evaluating treatments. However, they have key limitations, such as behavioral phenotype, genetic differences with humans, and even different synaptic plasticity, which may influence therapy accuracy. Other murine models include humanized knock-in (KI) mice [22,61]. This model would be ideal for assessing phenotypic rescue following therapeutic interventions with human gene-targeted compounds. Other in vivo models, such as the Cynomolgus monkey, could help bridge these gaps, offering a more accurate behavioral assessment after administration of various pharmacological agents due to its genomic similarity to humans [10].

Human-induced pluripotent stem cell (hiPSC)-derived neurons are also a promising in vitro model, as they reliably recapitulate Munc18-1 haploinsufficiency and altered neuronal activity. These hiPSCs can harbor various mutations, which provides significant advantages, allowing researchers to test genotype effects and therapeutic efficacy on human neuronal network morphology and physiology [22]. Preclinical studies have already shown very promising results. Various gene therapy approaches may improve synaptic activity and reduce symptoms associated with the syndrome. This could transform the treatment of STXBP1 syndrome and improve the quality of life of patients, parents, and caregivers.

Possibly the greatest challenge for personalized therapy in STXBP1 syndrome is the lack of genotype–phenotype correlation due to the high phenotypic heterogeneity caused by various known pathogenic variants. Some recurrent variants are known to possess specific phenotypic characteristics, but the baseline variability in clinical phenotypic features is so high that it prevents classification into discrete subgroups. Therefore, it will be essential to conduct genomic studies on variants using different methods to increase knowledge about STXBP1 syndrome’s clinical presentation [25,26]. In this regard, artificial intelligence and machine learning will be especially important, as they allow for the creation of predictors that identify and establish genotype-phenotype relationships, variant pathogenicity [69,74], and find biomarkers that facilitate response prediction to certain treatments. Finally, discovering and developing biomarkers for STXBP1 epileptic encephalopathy will be crucial for early disease detection, predicting possible progression, and evaluating potential response to specific treatments. Although ambitious, the first steps in this direction are already being taken. Studies analyzing and identifying EEG and body fluid signatures will be important in the near future for monitoring treatment responses [21].

## 8. Conclusions

This review provides a comprehensive examination of the challenges and recent advancements in STXBP1 syndrome research, a severe form of pediatric epileptic encephalopathy. The syndrome, caused by mutations in the STXBP1 gene, leads to significant clinical manifestations including seizures, motor impairments, and cognitive disabilities. This review highlights the critical role of the Munc18-1 protein in synaptic transmission, elucidating how haploinsufficiency and other dysfunctions resulting from pathogenic mutations underlie the disease mechanism. Gene therapy, particularly the use of antisense oligonucleotides (ASOs), emerges as a promising strategy to restore STXBP1 functionality, although current approaches still face technical and ethical challenges.

Experimental models, both in vivo and in vitro, have proven invaluable for studying disease mechanisms and testing potential therapies. Notably, in vitro models such as induced pluripotent stem cells (iPSCs) play a crucial role in personalizing treatments. Concurrently, integrated genomic studies and artificial intelligence are facilitating the exploration of genotype–phenotype correlations and the discovery of biomarkers for early diagnosis and therapeutic prediction.

Despite substantial progress over the past five years, significant hurdles remain. High phenotypic variability, low genotype–phenotype correlation, and a lack of clinical biomarkers continue to limit the development of effective treatments. To overcome these challenges, there is a pressing need for prospective studies, personalized approaches, and multidisciplinary collaboration.

Looking ahead, key areas for future research include elucidating unknown molecular mechanisms of STXBP1 pathology, identifying genes involved in STXBP1 regulation, developing more accurate animal models, improving genotype–phenotype correlations, and discovering and validating clinically relevant biomarkers. These efforts will be crucial in advancing our understanding of STXBP1 syndrome and translating scientific discoveries into effective treatments that can meaningfully improve the lives of patients and their families.

## Figures and Tables

**Figure 1 biotech-14-00011-f001:**
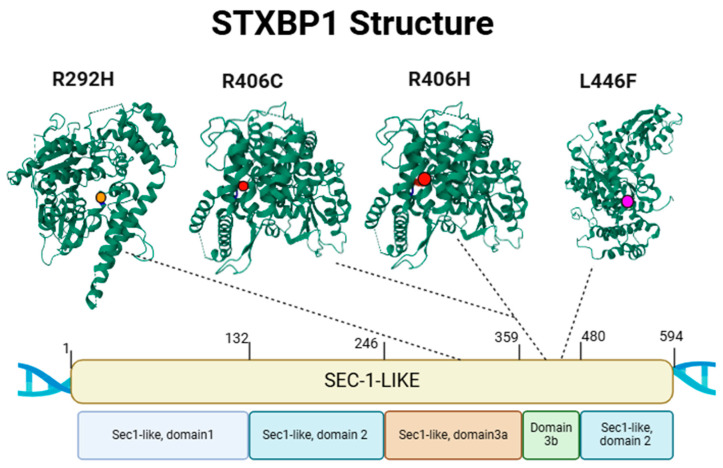
Structure of the STXBP1 gene and location of common pathogenic variants.

**Figure 2 biotech-14-00011-f002:**
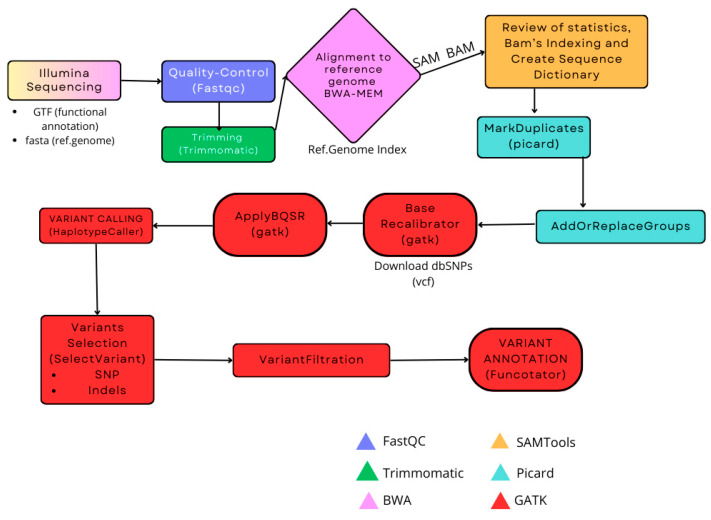
Overview of the Genomic Variant Detection Analysis conducted with STXBP1 patients. The workflow will begin with obtaining short DNA reads from patients, followed by quality control and trimming to filter out noise. Reads will then be mapped against a curated reference genome from ENSEMBL, with alignment statistics reviewed. The coordinate-sorted BAM file will be indexed, and the reference genome is indexed for alignment purposes. Duplicates will be marked for removal, and the read group information will be added. Base quality recalibration will be performed, followed by applying the recalibrated model. Variant calling will then be executed, with variants categorized by type. Standard GATK filters will be applied, and functional annotation will be conducted using Funcotator for interpretation, after downloading relevant databases. The bottom right corner shows color-coded programs or software used in each analysis step, including FastQC, Trimmomatic, BWA-MEM, SAMtools, Picard, and GATK. This comprehensive pipeline will enable the identification and characterization of genetic variants in STXBP1 syndrome patients, facilitating genotype–phenotype correlations and personalized medicine approaches [67,68].

**Figure 3 biotech-14-00011-f003:**
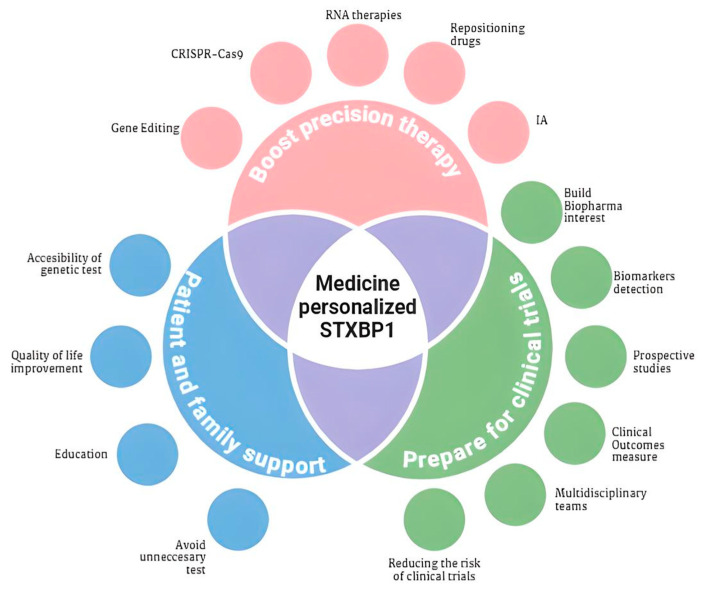
Key factors for advancing personalized medicine in STXBP1 syndrome. The diagram illustrates three fundamental pillars essential for developing effective precision therapy: driving new precision therapies, preparing clinical therapy, and providing support for patients and families affected by STXBP1 syndrome. Each pillar is further broken down into specific components crucial for achieving the overall goal of personalized treatment. The interconnected nature of these elements underscores the comprehensive approach needed to address the complex challenges of STXBP1 syndrome. This figure was created by the author using BioRender, inspired by the work of Goss et al. [21], and adapted to highlight the specific needs and opportunities in STXBP1 syndrome research and treatment.

## Data Availability

Data generated and analyzed in this study were derived from human samples and contain sensitive information. Due to privacy and data protection regulations, data’s patients cannot be shared publicly. Access to the data may be considered upon reasonable request and subject to ethical and legal approvals. Request can be directed to the authors.

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
