# Peer review of "STXBP1 Syndrome: Biotechnological Advances, Challenges, and Perspectives in Gene Therapy, Experimental Models, and Translational Research"

_biotech, 2025, doi:10.3390/biotech14010011_

Round 1

Reviewer 1 Report

Comments and Suggestions for Authors

Authors have presented the review on status of research on STXBP1 syndrome which is a epileptic encepalopathy condition, has overlap with other epileptic encephalopathies and its cases are likely under-reported. Given it’s a complex neurological condition, more research is required in the area and review on this topic is appropriate and serves a good purpose for such rare diseases/syndromes. However, there are clear issues in the review and it needs a significant improvement before being accepted for a publication. Below are my comments:

-       Given this is a review authors should have provided more background on similar encepalopathies which can be find elsewhere such as OMIM database.

-       Authors missed to provide the protein identifier for STXBP1/Munc18-1 in the review, they should provide the UniProt accession so that it’s easier to find which protein is being discussed and where different mutations are located.

-       Is the 2016 Neurology paper the first report of this syndrome? It’s important to cite the original paper where this condition was first identified.

-       “Additionally, we provide a detailed bioinformatic analysis of a Spanish cohort of 41 individuals with STXBP1-related disorders, offering insights into specific mutations and their biological implications.” -- Abstract says this but I do not see any analyses done in the review. Off-course authors seem to have access to a rich cohort of Spanish patients but they are only presenting a strategy and there is no real analyses presented in review. The section 5 should be presented as a “Potential strategy for Bioinformatic analysis of genomic data from our cohort of patients”

-       Page 2, line 84 – “There STXBP1 syndrome is primarily caused by haploinsufficiency, where mutations lead to reduced levels of functional”

should be

“The STXBP1 syndrome is primarily caused by haploinsufficiency, where mutations lead to reduced levels of functional”

-       Authors should carefully check full review again and correct any such Grammatical/typo issues.

-       Figure 1 - Structure of the STXBP1 gene –  Figure is of poor quality (not sure if this is in PDF only as I don't see the raw figure images) and possibly has a mistake. Domain 3a/3b is flanked by Domain 2 – this seems incorrect? How are these domains described as they are not present in PFAM/INTERPO databases. It’ bit weird to see such non-systematic numbering in protein domains. The linear cartoon for domains should specify the domain boundaries (amino acid numbering) for clarity. Authors should choose a good 3D visualization program Chimera-X/PyMol and rendered 3D representation for the variants they mentioned.

-       Section 3. Gene and Cell therapy: approaches and challenges. – Did authors review current clinical trials data for this disease in clinicaltrials.gov database?

-       Page 4, L156 - “These synthetic RNA molecules that mimic cell’s microRNAs” –  Add a reference to this statement.

-       P4, L175 - “to elevate cell levels of functional protein by ASOs” should be “to elevate cellular levels of functional protein by ASOs”

-       P6, L284 – “This issue is observed in the SYNGAP1 example.”  - authors should add reference to this statement.

-       Overall, the statements in section #5 (Bioinformatic analysis of genomic data from our cohort of patients) should be refined for the tense usgae, for instance, “The bottom right corner will show color-coded programs or software..” What do they mean by mentioning “will show” here. It should state that “colors represent” something meaningful which are programs/software in this case. The whole section 5 needs to be adapted to make it consistent as this is a proposed strategy.

-       Figure 3 – Here also, resolution is not optimal for the figure in pdf. Maybe authors should have prepared a high resolution image. The legend also has issue of not linking the exact reference next to “Goss et al., 2017” which is probably reference #18 in the current bibliography.

-       Figure 3 legend – “the reference genome will be indexed” should be changed to “the reference genome will be indexed for alignment purposes”

Author Response

Dear Reviewer, 

The response to your proposed correction is attached in the Word document.

Thank you for your support and consideration throughout this process. We look forward to your feedback and the next steps in the editorial process.

Reviewer 2 Report

Comments and Suggestions for Authors

MS entitled ‘Biotechnological challenges in STXBP1 syndrome research: Gene and Cell Therapy, Experimental Models and Future Clinical Perspectives’ has addressed the challenges in STXBP1 syndrome research along with utility of Gene and Cell Therapy, Experimental Models and also reviewed Future Clinical Perspectives. Authors have comprehensively reviewed STXBP1 syndrome research, focusing on: pathogenic mechanisms and mutation types in STXBP1 syndrome; advanced gene therapy approaches and challenges; experimental models for STXBP1 research; bioinformatics analysis of genomic data from a Spanish patient cohort; clinical and translational perspectives and future research directions and challenges. Somehow, authors have not presented information in a systematic way while covering current knowledge, highlighting recent advancements, and identifying critical areas for future investigation in STXBP1 syndrome.

Following are the specific comments regarding the study.

Remark 1: What is direct meaning of STXBP1 syndrome? It is to be initiating point of description.

Remark 1: The authors have put their efforts to identify key challenges but in MS they have presented wide spectrum information. As authors are hoping that by this they will contribute to the ongoing efforts to improve diagnosis, treatment, and quality of life for individuals affected by this complex neurological disorder. Somehow, they lack pinpointing the important understanding with the specific field as per their title of MS.   

Remark 2: Improve English language and grammar. Page 7-9 line 329-375 Details are given in future tense which need to be rectify or reanalyze.

Remark 3: MS have missed some necessary citation/reference like ‘Studies using the CRISPR/Cas9 mechanism have being developed in order to understand the pathophysiology of the syndrome and to develop a targeted therapy’. Page 4, line 135

Remarks 4: In figures, clarity to be further enhanced to make it understandable.

Remarks 5: MS, to be made systematic as it has provided information in patches which as repetition as well.

Comments on the Quality of English Language

Improve English language and grammar. Page 7-9 line 329-375 Details are given in future tense which need to be rectify or reanalyze.

Author Response

(The authors gave the same response as above.)

Round 2

Reviewer 1 Report

Comments and Suggestions for Authors

#Authors have made changes to the manuscript based on the report, however, all comments haven’t been addressed in entirety, these include:

“Figure 1 - Structure of the STXBP1 gene –  Figure is of poor quality (not sure if this is in PDF only as I don't see the raw figure images) and possibly has a mistake. Domain 3a/3b is flanked by Domain 2 – this seems incorrect? How are these domains described as they are not present in PFAM/INTERPO databases. It’ bit weird to see such non-systematic numbering in protein domains. The linear cartoon for domains should specify the domain boundaries (amino acid numbering) for clarity. Authors should choose a good 3D visualization program Chimera-X/PyMol and rendered 3D representation for the variants they mentioned.”

-       The figure in text is still the old version and still lacks clarity. Authors cited another paper to explain the domain numbering issue, however, this is not enough. They should explain why the naming is being done in a non-consequential manner. Is it some kind of typo in the previous paper (Uddin et al. [23,28]) which is becoming a legacy now? A clear explanation or appropriate changes would alleviate the concern.

#Authors have cited OMIM but missed adding the reference for the resource. They should cite it appropriately.

#Authors have updated section 5, however, there are still inconsistencies given this is a future plan:

“In this section, a comprehensive bioinformatic analysis of genomic data from a cohort of Spanish patients with STXBP1 syndrome is presented.“

should be

“In this section, a comprehensive plan for bioinformatic analysis of genomic data from a cohort of Spanish patients with STXBP1 syndrome is presented.“

---

“Bioinformatic processing of short-read WGS data is conducted through the Picard”

to

“Bioinformatic processing of short-read WGS data will be conducted through the Picard”

 ---

“To eliminate most noise, they are processed with Trimmomatic v.0.39, a Java-based tool (Java v17.0.6), performing quality- based trimming and removing Nextera adapters. Cleaned reads are subjected”

to

“To eliminate most noise, they will be processed with Trimmomatic v.0.39, a Java-based tool (Java v17.0.6), performing quality based trimming and removing Nextera adapters. Cleaned reads will be subjected”

---

“The reads are then sorted by coordinates to streamline further analysis, and alignment success is verified.” 

to

“The reads will be then sorted by coordinates to streamline further analysis, and alignment success is verified.”

--

#Similarly remaining part of section 5 and legend should also be revised as per the future planning instead of using the present tense.

Author Response

Dear Reviewer,

The responses to your suggestions and requests for improvements are provided in the attached Word document.

Thank you very much for your consideration and support.

On behalf of all authors,
Silvestre Ruano Rodríguez.

Round 3

Reviewer 1 Report

Comments and Suggestions for Authors

Authors have improved the manuscript based on previous comments. Now, the structure in Fig. 1 has been rendered using Chimera-X, however, the tool is not cited in text and reference for the same is missing in bibliography. Real STXBP1 structure domain naming clarifies potential nomenclature related misunderstanding of the domains. The font size of the domains in figure differs, for instance, Sec1-like domain 2 font is different and should be fixed. Authors have now appropriately added the PDB id as well though reference for the PDB database is also missing. Both references [(1) PDB database (2) Chimera-X] should be cited and added in the bibliography.

Author Response

Dear Reviewer,

I sincerely appreciate the time and effort you dedicated to reviewing my manuscript. Your insightful comments and suggestions have been invaluable in improving the quality of the work.

Please find my detailed response in the attached Word file.

Best regards,
Silvestre Ruano.
